# Theoretical Investigation on the Catalytic Effect and Mechanism of Pure and Cu−Doped SBA−15 Molecular Sieves on the Decomposition of Dimethyl Sulfoxide

**Haohai Xia** [1,2,3]**, Xianglong Meng** [1,2]**, Xingchao Jiang** [4]**, Lilin Lu** [5] **and Yanqun Wang** [3,*]

1 State Key Laboraory of Shale Oil and Gas Enrichment Mechanisms and Effective Development, Beijing 100083, China
2 State Center for Research and Development of Oil Shale Exploitation, Beijing 100083, China
3 College of Chemistry and Environmental Engineering, Yangtze University, Jingzhou 434023, China
4 College of Resources and Environment, Yangtze University, Wuhan 430100, China
5 State Key Laboratory of Refractories and Metallurgy, Wuhan University of Science and Technology, Wuhan 430081, China
* Correspondence: qunqunlucky@whu.edu.cn; Tel.: +86-716-8060-650

**Abstract:** The interaction mechanism between oil shale and catalyst is very important for the design and synthesis of related catalysis. In this work, dimethyl sulfoxide (DMSO) serves as a model molecule for organic sulfur compounds in oil shale to explore the catalytic effect and mechanism of the pure and transition metal Cu−doped SBA−15 molecular sieves regarding the decomposition of organic sulfur compounds in oil shale using the density functional theory (DFT) method. It is found that DMSO adsorption on both surfaces is primarily attributed to hydrogen bonding or the interaction between the S and O moieties within the molecule and the surface Cu atoms. The adsorption energies on both surfaces are indistinguishable; however, the Cu−doped SBA−15 shows enhanced catalytic activity in dissociation reactions. The Gibbs free energy changes for both possible reaction pathways of DMSO breaking C−S bonds on the pure SBA−15 surface are positive, and the activation energy barriers are as high as ~75 kcal/mol, indicating that the dissociation of C−S bonds in DMSO is unlikely to occur on this surface. In contrast, the Gibbs free energy change for the same reaction on the Cu−doped SBA−15 surface is negative, and the energy barrier is reduced by ~40 kcal/mol compared to that on the pure SBA−15 surface. Furthermore, the resulting methyl group is more likely to bond with the bridging oxygen atom. In addition, our research proposes that the dissociation of the C−H and C−S bonds of DMSO on the Cu−doped SBA−15 surface was competitive. These findings provide theoretical guidance for the development of highly efficient catalysts intended for the pyrolysis of oil shale under appropriate conditions.

**Keywords:** oil shale; SBA−15 molecular sieve; catalytic cracking; dimethyl sulfoxide; density functional theory

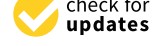



## 1. Introduction

Oil shale, as an unconventional oil and gas resource, is a non-renewable fossil energy similar to petroleum, natural gas, and coal. Due to its abundant resources and feasibility of development and utilization, it has been identified as an important substitute energy source for the 21st century and has become a strategic resource for the sustainable development of human society [1].

Liquid shale oil extraction from oil shale can only be achieved by pyrolysis. However, under non-catalytic conditions, the pyrolysis conversion rate of oil shale is relatively low (~20% [2,3]), which adversely affects the quality of shale oil, resulting in unstable oil and a high viscosity. To improve the pyrolysis conversion rate and the quality of shale oil and gas, it is necessary to investigate the catalytic pyrolysis of oil shale [4,5]. Molecular sieves, due to

their complex and diverse pore structure, high specific surface area, and acidity, can be used to selectively regulate the pyrolysis process [6], making them a commonly used catalyst in the petrochemical industry. Among the molecular sieve materials, SBA−15, an amorphous $SiO_2$ substrate, has demonstrated excellent adsorption and substance transfer properties in various applications, such as drug delivery [7], gas capture [8], substance separation [9], etc., owing to its tunable pore size within the range of 5–30 nm. Its controllable pore size range allows the entry and retention of large molecules such as kerogen and bitumen in oil shale for reaction; their large surface area may also lead to the high dispersion of active species and consequent high catalytic activity [10]. SBA−15's unique physical and chemical properties have drawn considerable attention from researchers. Researchers have applied a clin/SBA−15 molecular sieve to subcritical in situ mining technology to increase the H/C ratio and decrease the O/C ratio [11]. Additionally, studies have shown that the surface modification of SBA−15 molecular sieves by introducing catalytic active sites can enhance their catalytic activity. Doped Co and Ni into SBA−15 for methane catalytic reforming resulted in a higher conversion rate of methane and carbon dioxide in the catalytic system [12]. Notably, doped Al into SBA−15 catalysts for the in situ upgrading of shale oil effectively reduced the activation energy of organic matter conversion in shale and reduced energy consumption [13].

Although the introduction of active centers has been shown to improve the catalytic performance of SBA−15, the complex nature of the pyrolysis process means that the underlying mechanisms of oil shale pyrolysis under catalyst conditions remain poorly understood, while quantum chemical calculations hold considerable potential for providing insights into the microscopic nature of heat conversion in fossil fuels. Various quantum chemical calculations and infrared studies have shown that copper-modified molecular sieves can activate double and triple bonds in acetylene, olefins, acetone, and benzene [14]. Recent density functional theory studies have shown that $Cu^{2+}$ has the highest catalytic activity in the hydrothermal degradation of heavy oil. Among several transition metal ions, the catalytic activity sequence is $Cu^{2+} >> Co^{2+} \approx Ni^{2+} > Fe^{2+}$. As such, it is expected that the introduction of copper into SBA−15 will enhance its catalytic activity in the pyrolysis of oil shale. Nevertheless, the catalytic effect of Cu−doped SBA−15 and underlying mechanisms for the catalytic performance of SBA−15 have not yet been elucidated at the molecular level. Most of the existing literature based on quantum chemical calculations has focused on the structure and thermal decomposition of oil shale, with little attention being paid to the pyrolysis mechanism of oil shale. For example, Wang et al. [15] employed experimental results and density functional theory calculations to obtain the three-dimensional structure and possible carbon skeleton isomers of Huadian oil shale kerogen, yielding both the overall body structure and molecular information of oil shale. Ye et al. [16] utilized the transition state theory of quantum chemistry to investigate the thermochemical reaction between organic carbon and H radical during oil shale retorting. By constructing corresponding multiple reaction paths using a reasonably simplified direct carbon chain model, they clarified the mechanism of the thermochemical reaction.

Against this backdrop, the present study aims to investigate the mechanisms underlying the catalytic pyrolysis of oil shale using both pure and Cu−doped SBA−15, and to use these findings to inform the design and development of in situ catalytic pyrolysis catalysts under specific conditions. To achieve this goal, a series of calculations were performed, including the adsorption configurations, adsorption energies, and electronic structures of oil shale model molecules on the surfaces of pure and Cu−doped SBA−15. Additionally, the energy barriers and Gibbs free energy changes associated with the dissociation reactions of C−S and C−H bonds in model molecules on different catalyst surfaces were also determined.

## 2. Computational Method

Density functional theory (DFT) is a kind of ab initio theory, which is a purely mathematical calculation based on some basic principles of classical quantum mechanics and is established for the operation of a multi-electron system without any other empirical

constants. All the DFT calculations were performed using the DMol$^3$ program [17], employing the generalized gradient approximation (GGA) [18] and Perdew–Burke–Ernzerhof (PBE) [19] functional, with the incorporation of Grimme's DFT-D method for the correction of van der Waals forces [20]. Copper atoms were treated using the DFT semi-core pseudopotential (DSPP) [21], and a double numerical all-electron basis set (DND) [22] with d polarization functions was applied to all heavy atoms. The structure was relaxed using the Broyden–Fletcher–Goldfarb–Shanno (BFGS) method [23], and the integration in the Brillouin zone was performed on a $3 \times 3 \times 1$ k point Monkhorst–Pack grid. The real space cutoff value was set to 5.3 Å, and a thermal tailing effect of 0.005 au was introduced to accelerate convergence. The convergence criteria were set to a total energy change of less than $2 \times 10^{-5}$ eV between neighboring steps, an energy gradient of less than 0.004 eV/Å, and an atomic displacement of less than 0.005 Å during the optimization process.

Synchronous transit methods (LST/QST) [24] were employed to obtain the initial approximations of transition states, which were subsequently fine-tuned by means of eigenvector following optimization. The optimized structures were subsequently subjected to harmonic vibrational frequency calculations, whereby the mass-weighted second-derivative matrix was diagonalized utilizing two-sided finite differences featuring a displacement step of 0.01 Å and Hessian sets of relaxed atoms.

The expression used to calculate the adsorption energy was $E_{ads} = E_{mol} + E_{surf} - E_{(mol+surf)}$, where $E_{mol}$ represents the energy of a single molecule of dimethyl sulfoxide (DMSO) within a 20 Å $\times$ 20 Å $\times$ 20 Å periodic cubic unit, $E_{surf}$ represents the energy of the catalyst surface, and $E_{(mol+surf)}$ represents the total energy of the system when the DMSO molecule is on the catalyst surface. It is established that positive adsorption energy indicates a stable adsorbate/surface system [25].

## 3. Results and Discussion

### 3.1. Structures of the Pure and Cu−Doped SBA−15 Surfaces

A periodic film of hydroxylated amorphous silica was employed, as depicted in Figure 1. The structures were taken from prior works, which were based on amorphous SiO$_2$ (111) surface and were obtained by moving the thin layer position in the (111) direction. The details of their structural characterization are published in these works [26,27]. Furthermore, Wang et al. [28] conducted a comprehensive study of the geometries of silica rings in SBA−15 and identified six-membered and six-membered rings as the principal structural components. The Si atom, which is substituted by the Cu atom, was surrounded by six-membered and five-membered rings and was attached by one hydroxyl. The unit cell dimensions are 12.8 $\times$ 17.0 $\times$ 23.7 Å$^3$. The total number of atoms is 94 (CuSi$_{17}$O$_{49}$H$_{27}$).

As illustrated in Figure 1, the three bridging oxygen atoms connected to Si or Cu are denoted "O$_b$" and numbered in a clockwise fashion. The charge transfer within the system can influence both the atomic chemical reactivity and bond lengths. To investigate the influence of Cu doping on the structure features of SBA−15, a comparison analysis of Si−O bond lengths pre- and post-doping with Cu was performed, as shown in Table 1. Additionally, a Mulliken analysis was conducted to explore the charge transfer that occurs between the Cu atoms and neighboring oxygen atoms, as delineated in Table 1.

According to Table 1, in contrast to the Si−O$_b$ bond length, following the introduction of Cu into the SBA−15, the Cu−O$_b$ bond length increases from 1.625 Å, 1.639 Å, and 1.621 Å on the pure SBA−15 surface to 1.808 Å, 1.795 Å, and 1.830 Å, respectively. The elongated bonds provide additional space for molecular adsorption and reaction. Furthermore, a Mulliken charges analysis shows that after Cu doping, the charge of the bridge oxygen atom O$_b$ connected to the Cu atom changes from $-0.884$ |e|, $-0.877$ |e|, and $-0.902$ |e| to $-0.665$ |e|, $-0.661$ |e|, and $-0.724$ |e|, respectively, indicating that the charge in the O$_b$ atom is transferred to the Cu atom, leading to a diminution of the electrical quantity of the O$_b$ atom.

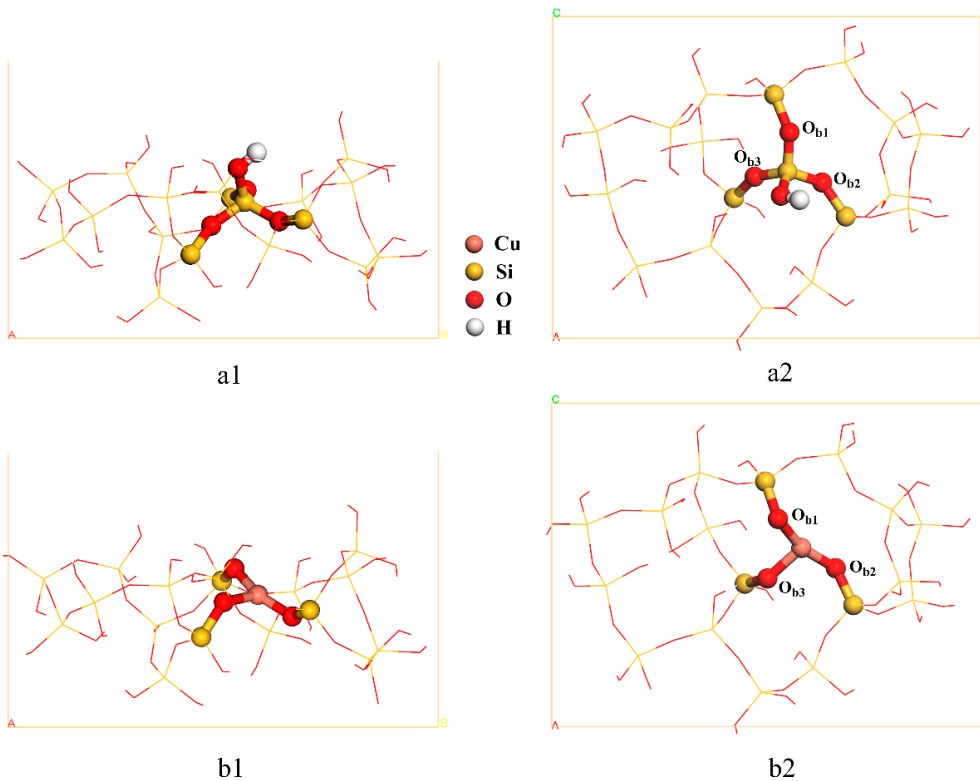

**Figure 1.** Structures of the pure SBA−15 and Cu−doped SBA−15 models used in this work: (**a1,a2**) represent the side view and top view of the pure SBA−15, respectively. (**b1,b2**) represent side view and top view of the Cu−doped SBA−15, respectively.

**Table 1.** X−$O_b$ (X is Si or Cu atom) bond length and Mulliken charge of $O_b$ atom in SBA−15 before and after Cu doping.

| Bond Type | Bond Length (Å) | | Atom | Mulliken (∣e∣) | |
|---|---|---|---|---|---|
| | **Pure-SBA−15** | **Cu-SBA−15** | | **Pure-SBA−15** | **Cu-SBA−15** |
| D (X−$O_{b1}$) | 1.625 | 1.808 | $O_{b1}$ | −0.884 | −0.665 |
| D (X−$O_{b2}$) | 1.639 | 1.795 | $O_{b2}$ | −0.877 | −0.661 |
| D (X−$O_{b3}$) | 1.621 | 1.830 | $O_{b3}$ | −0.902 | −0.724 |

### *3.2. The Adsorption of DMSO on the Pure and Cu−Doped SBA−15 Surfaces*

3.2.1. Adsorption Configuration and Energy for DMSO Adsorbed on the Pure SBA−15 Surface

Kang et al. [29] found that the rupture of heteroatomic bonds within kerogen in oil shale can initiate a network of chain cleavage reactions within the kerogen, resulting in an improvement in the kerogen cleavage rate. It is noteworthy that the principal heteroatoms present within kerogen are the nitrogen (N) and sulfur (S) atoms, with the latter being predominantly found in the form of sulfone and sulfoxide. Consequently, DMSO ($C_2H_6OS$) was chosen as the model molecule of the S element in oil shale in this work.

Figure 2 depicts the most stable adsorption structure and the associated adsorption energy of the interaction between DMSO and the pure SBA−15 surface. To simplify the description, the notation "pure-SBA15-DS" is employed to denote the most stable adsorption structure of DMSO on the pure SBA−15 surface. As shown in Figure 2, DMSO mainly interacts with the surface of the molecular sieve via its O or H atoms, and the adsorption energy is 32.6 kcal/mol. The oxygen atom of DMSO forms a hydrogen bond with the hydrogen atom in the hydroxyl group connected to the silicon atoms within the five-membered and six-membered rings, and the distance between the oxygen and hydrogen atoms measures 1.631 Å. Furthermore, the hydrogen atoms within the DMSO

molecule are attracted by the nearby twin silanol [30] and bridging oxygen atoms, with atomic separations of 2.632 Å and 2.896 Å, respectively.

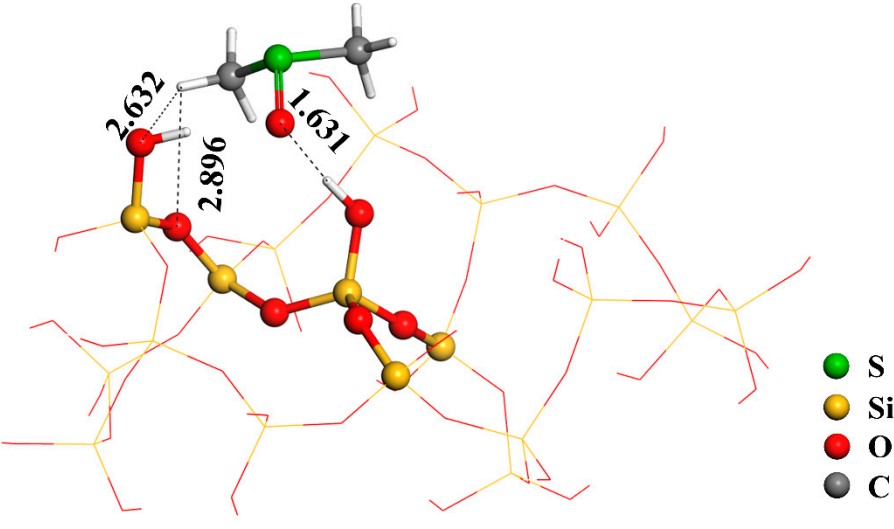

pure-SBA15-DS ( 32.6 kcal/mol )

**Figure 2.** The most stable adsorption structure (bond length: Å) and energy (kcal/mol) for DMSO absorbed on the pure SBA−15 surface.

### 3.2.2. Adsorption Configuration and Energy for DMSO Adsorbed on the Cu−Doped SBA−15 Surface

According to the study by Lu et al. [31], the node of the Si−O five-membered ring and six-membered ring in SBA−15 is the most active site. They studied the complex reaction mechanism of ethanol to 1,3-butadiene by doping Zr atoms at this site [31]. In this work, the Si atom has been replaced on the pure SBA−15 surface at this site with Cu atoms, thus constructing a Cu−doped SBA−15 molecular sieve surface. The introduction of Cu atoms has resulted in an increase in the number of Lewis acid sites on the surface. By investigating the various orientations and adsorption sites of DMSO molecules on the Cu−doped SBA−15 surface, the adsorption structures of DMSO on this surface were obtained. Figure 3 illustrates the representative adsorption structures of DMSO on the Cu−doped SBA−15 surface. For the sake of brevity, "Cu-SBA15-DS" was used to refer to the adsorption structure throughout the following exposition.

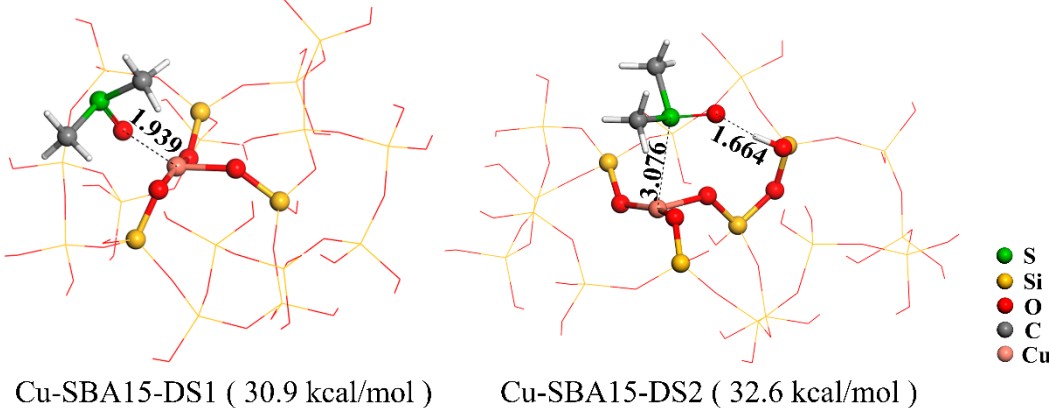

Cu-SBA15-DS1 ( 30.9 kcal/mol )  Cu-SBA15-DS2 ( 32.6 kcal/mol )

**Figure 3.** Representative adsorption structures (bond length: Å) and energies (kcal/mol) of DMSO on the Cu−doped SBA−15 surface.

From Figure 3, it is evident that there are two distinct types of adsorption structures of DMSO on the Cu−doped SBA−15 surface. These structures involve the interaction of the oxygen and sulfur atoms within the DMSO molecule with the surface Cu atoms. Notably, the adsorption structure in which the sulfur atom interacts with the surface Cu atom is more stable, with an adsorption energy of 32.6 kcal/mol. This value is equivalent to the adsorption energy of DMSO on the pure SBA−15 surface, thereby indicating that the introduction of Cu atoms has had a negligible impact on the adsorption behavior of DMSO on the SBA−15 surface.

### 3.2.3. Density of States

To provide further insight into the interactions between DMSO and the pure and the Cu−doped SBA−15 surfaces, the electronic structures for the most stable adsorption configurations (i.e., pure-SBA15-DS in Figure 2 and Cu-SBA15-DS2 in Figure 3) were investigated. Figure 4 shows the partial densities of states (PDOS) of the most stable configurations before and after DMSO adsorption. More specifically, as shown in Figure 4a, the p-PDOS of O in DMSO shifts to a lower energy compared with the initial configuration. In addition, the O s-PDOS interacts with the H s-PDOS at −9.15 eV, and the O p-PDOS interacts with the H s-PDOS at −9.15 and −6.65 eV on the pure SBA−15 surface after adsorption, which is attributed to the formation of hydrogen bonds and van der Waals forces. In the case of the Cu−doped SBA−15 surface, both the s-PDOS and p-PDOS of the S atom in DMSO show a trend of shifting to a lower energy compared with the initial configuration. For example, the peak value of S s-PDOS decreased at −19.7 eV and −9.1 eV, and the peak of S p-PDOS decreased at −4.1 eV and −2.9 eV. In addition, the S s-PDOS interacts with the Cu s-PDOS by state overlap at −2.84 eV, and the S p-PDOS interacts with the Cu p-PDOS by state overlap at −1.93 eV, as can be seen in Figure 4b; these energies correspond to the interaction of S and Cu atoms present in Cu-SBA15-DS2.

### *3.3. The Dissociation of DMSO on the Pure and Cu−Doped SBA−15 Surfaces*

### 3.3.1. The Dissociation of DMSO on the Pure SBA−15 Surface

In this section, the catalytic cracking process of DMSO on the pure SBA−15 surface was investigated, using the most stable adsorption structure of DMSO on the pure SBA−15 surface (i.e., pure-SBA15-DS in Figure 2) as the reactant. DMSO comprises three types of chemical bonds, namely, C−H, C−O, and C=S bonds. In view of the fact that the bond energy of double bonds is greater than that of single bonds, the focus is primarily on the breaking of C−H and C−S bonds. Structural optimization reveals that the optimization process of C−H bond breaking yielded no dissociation product. Moreover, research has shown [32] that the bond energy of C−H and C−S bonds is 96–99 kcal/mol and 66 kcal/mol, respectively, signifying that C−S bonds are more prone to cleavage than C−H bonds. As a consequence, the breaking mechanism of C−S bonds in DMSO molecules on the pure SBA−15 surface has been examined in greater detail, and Figure 5 displays the dissociation structure of DMSO on the pure SBA−15 surface in which C−S bonds are broken. For the sake of convenience, the term "DS" is employed to designate the dissociation structure.

As shown in Figure 5, following the breaking of the C−S bond in DMSO, the dissociated methyl group binds with oxygen atoms at varied sites on the surface of pure SBA−15. These dissociation structures can be distinctly classified into two categories, namely DS-1 and DS-2, which are exhibited in Figure 5. The former denotes the attachment of the methyl group to the surface hydroxyl O atom, while the latter involves its binding to the surface $O_b$ atom. Simultaneously, it is noteworthy that the transfer of the hydrogen atom from the surface hydroxyl group to the O atom of DMSO leads to the formation of the $CH_3SOH$ molecule.

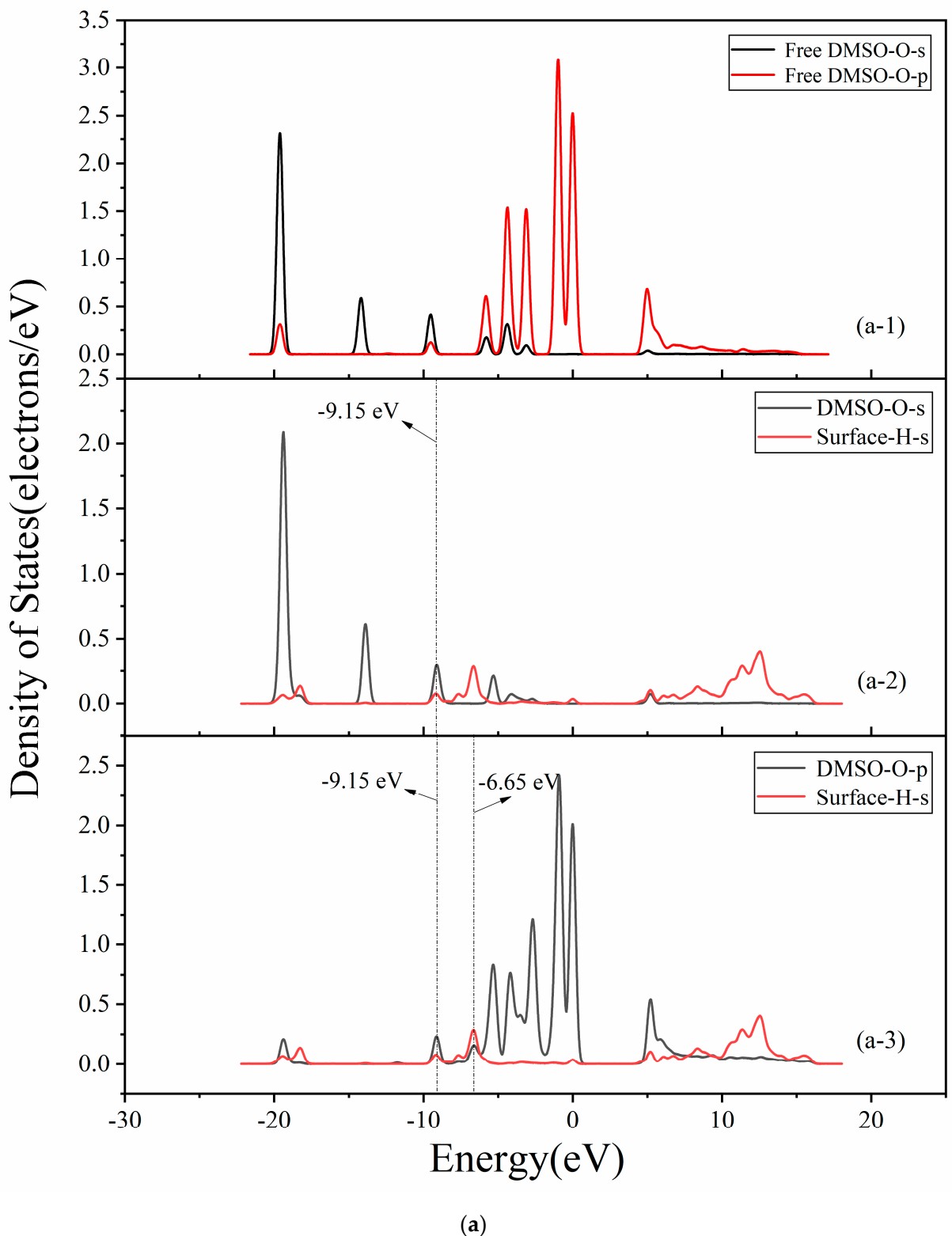

**Figure 4.** *Cont.*

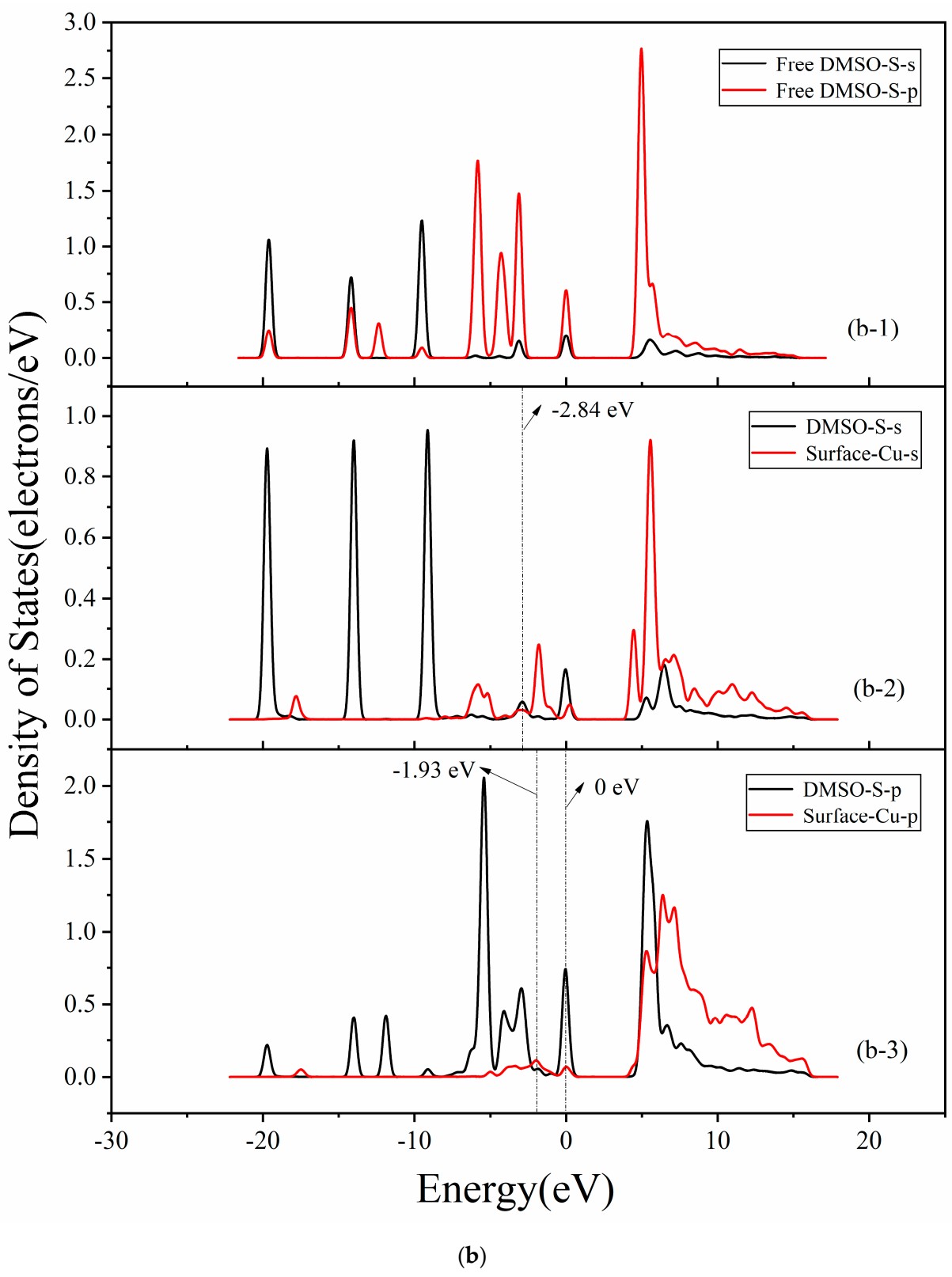

**(b)**

**Figure 4.** The PDOS of the most stable configurations for DMSO on the pure (**a**) and Cu−doped SBA−15 (**b**) surfaces before and after adsorption: (**a-1**) represents the O s- and p-PDOS before DMSO adsorption, (**a-2**) represents the O s-PDOS and H s-PDOS and (**a-3**) represents the O p-PDOS and H s-PDOS after DMSO adsorption. (**b-1**) represents the S s- and p-PDOS before DMSO adsorption, (**b-2**) represents the S s-PDOS and Cu s-PDOS and (**b-3**) represents the S p-PDOS and Cu p-PDOS after DMSO adsorption.

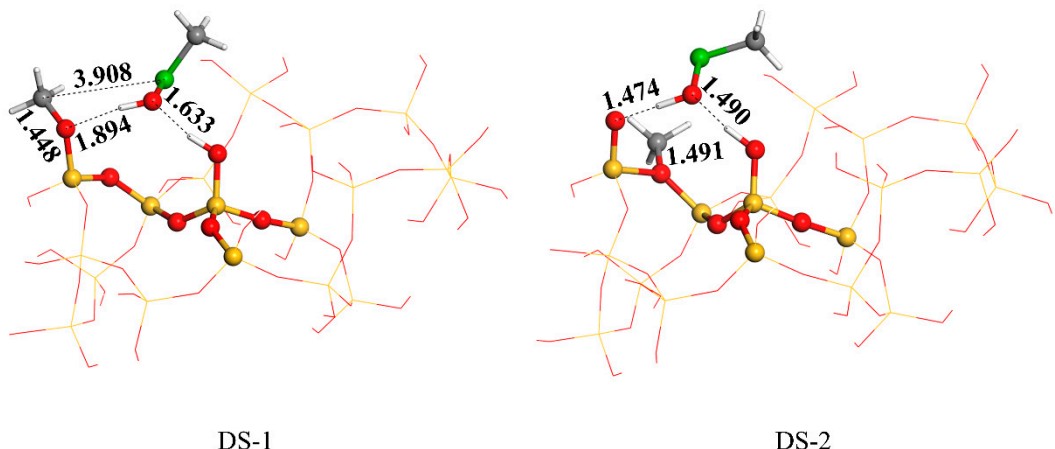

DS-1                                           DS-2

**Figure 5.** Structure of decomposition product for DMSO breaking C−S bond on the pure SBA−15 surface.

Table 2 presents the Gibbs free energy changes associated with the reaction pathways leading to the two dissociation products in Figure 5. It is evident that the C−S bond cleavage reaction of DMSO on the pure SBA−15 surface exhibits a positive Gibbs free energy change. Specifically, the generation of DS1 and DS2 requires overcoming energies of 14.7 and 56.7 kcal/mol, respectively, indicating that the C−S bond cleavage of DMSO on the pure SBA−15 surface cannot occur spontaneously. Moreover, transition state search for both reaction pathways revealed that the barriers to form DS-1 and DS-2 are 81.5 and 74.6 kcal/mol, respectively, which are high and do not easily occur at room temperature. Such outcomes signify that the dissociation reaction of DMSO on the pure SBA−15 surface is both thermodynamically and kinetically unfavorable. Therefore, from a chemical activity perspective, pure SBA−15 is not suitable as a reaction catalyst. This conclusion is consistent with the findings in the literature [33] which classify pure SBA−15 as an inert carrier material.

**Table 2.** Gibbs free energy variation $\Delta G$ and energy barrier $\Delta E$ of reaction paths for DMSO on the pure and Cu−doped SBA−15 surfaces.

| Paths | Surface | Product | $\Delta G$ (kcal/mol) | $\Delta E$ (kcal/mol) |
|---|---|---|---|---|
| $CH_3 \rightarrow O_b$ | pure-SBA−15 | DS1 | 14.7 | 81.5 |
| $CH_3 \rightarrow OH$ | pure-SBA−15 | DS2 | 56.7 | 74.6 |
| $H \rightarrow OH$ | Cu-SBA−15 | Cu-DS1 | 21.6 | 62.2 |
| $H \rightarrow O_b$ | Cu-SBA−15 | Cu-DS2 | 12.1 | 39.8 |
| $CH_3 \rightarrow OH$ | Cu-SBA−15 | Cu-DS3 | 6.1 | 63.8 |
| $CH_3 \rightarrow O_b$ | Cu-SBA−15 | Cu-DS4 | −16.4 | 31.1 |

### 3.3.2. The Dissociation of DMSO on the Cu−Doped SBA−15 Surface

This section focuses on the catalytic cleavage process of DMSO on the Cu−doped SBA−15 surface. The most stable adsorption structure of DMSO on the Cu−doped SBA−15 surface, i.e., Cu-SBA15-DS2 in Figure 3, was utilized as the reactant. The cleavage process involves the breaking of C−H and C−S bonds in DMSO. Figure 6 illustrates the typical dissociation products resulting from the cleavage of the C−H or C−S bond in DMSO on the Cu−doped SBA−15 surface, which are designated as "Cu-DS" for convenience. The energy profiles for each reaction process are depicted in Figure 7. The Gibbs free energy changes for each dissociation reaction are provided in Table 2.

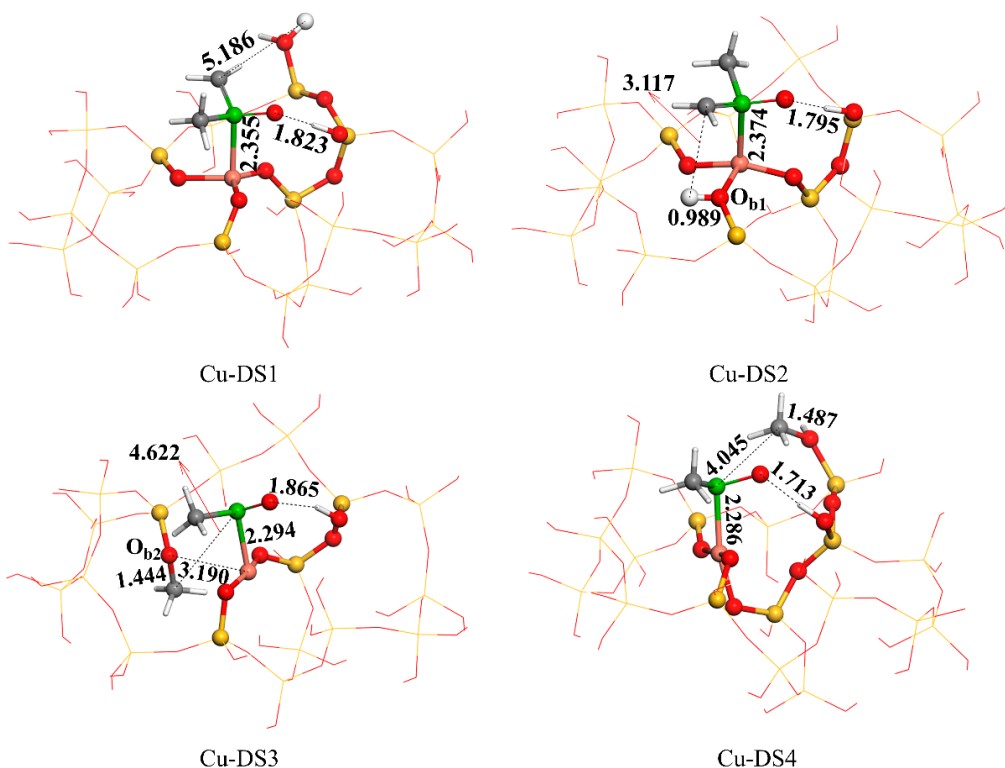

**Figure 6.** Structure of decomposition product for DMSO breaking C−H or C−S bond on the Cu−doped SBA−15 surface.

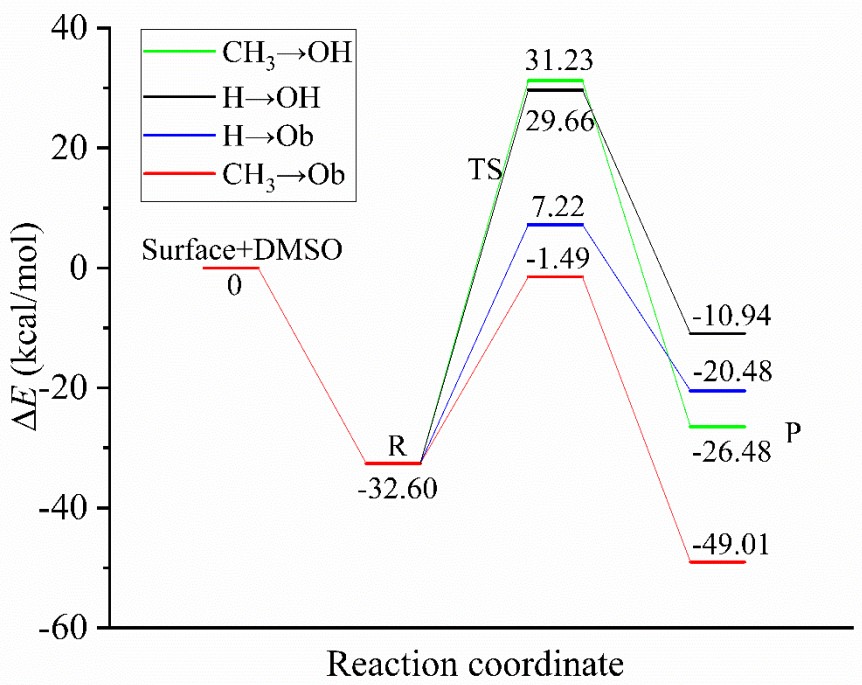

**Figure 7.** The potential energy diagram of DMSO decomposition on the Cu−doped SBA−15 surface.

As demonstrated in Figure 6, the Cu−doped SBA−15 surface yields four distinct types of decomposition products of DMSO. Specifically, Cu-DS1 and Cu-DS2 represent dissociative structures characterized by C−H bond cleavage and H atom binding to the surface hydroxyl O atom and surface bridge O atom, respectively. The Gibbs free energy changes in these two dissociation processes are both positive, as indicated in Table 2, with values of 21.6 and 12.1 kcal/mol, respectively. By searching the transition states of these

two dissociation processes, it has been determined that the energy barrier associated with the H atom binding to the hydroxyl O atom is notably high at 62.2 kcal/mol, thus rendering it a rare occurrence at room temperature. On the other hand, the energy barrier associated with the H atom binding to the bridge O atom is approximately 20 kcal/mol lower, with a value of 39.8 kcal/mol, as demonstrated in Figure 7.

Cu-DS3 and Cu-DS4, represented in Figure 6, correspond to the dissociation structure of the methyl group bonded with hydroxyl O or $O_b$ atoms on the surface after C−S bond break. The Gibbs free energy change associated with the dissociation process resulting in the formation of Cu-DS4 is −16.4 kcal/mol, which is negative, indicating the thermodynamic feasibility of the process. Upon investigation of the transition states of the two dissociation processes, the energy barrier associated with methyl binding to the hydroxyl O atom was found to be 63.8 kcal/mol, while that associated with methyl binding to the $O_b$ atom was approximately 30 kcal/mol lower, with a value of 31.1 kcal/mol, as demonstrated in Figure 7.

The above analysis demonstrates that the C−S bond cleavage of DMSO on the Cu−doped SBA−15 surface, with the methyl group binding to the surface $O_b$ atom, is a reaction pathway that is both thermodynamically and kinetically favorable. Furthermore, it has been demonstrated that Cu doping promotes the decomposition of DMSO compared with the reaction of DMSO on the pure SBA−15 surface.

## 4. Conclusions

The present study aims to investigate the interactions of the model organic sulfur molecule DMSO with the pure and Cu−doped SBA−15 zeolites, utilizing a periodic model and first-principles calculations based on density functional theory. The GGA/PBE functional and DND basis set were employed for these analyses, which included the examination of the adsorption and decomposition of DMSO on both surfaces. It was found that DMSO molecules were mainly adsorbed on two surfaces by hydrogen bonding and van der Waals interaction, and the adsorption energy of both was equal to 32.6 kcal/mol. On the pure SBA−15 surface, DMSO adsorption is primarily attributed to the interaction between the O and H moieties within the molecule and the H and O atoms present on the surface. Conversely, on the Cu−doped SBA−15 surface, the adsorption mechanism is largely influenced by the interaction between the S and O moieties within the molecule and the Cu atoms on the surface. Further studies on the dissociation reactions indicate that, on the pure SBA−15 surface, the DMSO molecule only produces dissociation products when the C−S bond is broken, while no C−H bond cleavage is observed. Moreover, for the C−S bond cleavage reaction on the pure SBA−15 surface, both the Gibbs free energy change and the energy barrier suggest that the dissociation process is difficult to proceed. For C−S bond breaking, accompanied by -$CH_3$ bonding to hydroxy oxygen atoms on the surface, the Gibbs free energy change and the energy barrier are 14.7 and 81.5 kcal/mol, respectively. For C−S bond breaking, accompanied by -$CH_3$ bonding to the bridging oxygen atom, the Gibbs free energy change and the energy barrier are 56.7 and 74.6 kcal/mol, respectively. In contrast, on the Cu−doped SBA−15 surface, the C−S bond cleavage of DMSO with the methyl group binding to the surface bridging oxygen atom is a reaction pathway that is both thermodynamically and kinetically favorable. The corresponding Gibbs free energy change is −16.4 kcal/mol, and the energy barrier is 31.1 kcal/mol, which is lower by ~40 kcal/mol than that on the pure SBA−15 surface, implying an improved catalytic effect for DMSO decomposition.

Notably, the energy barrier for the breaking of the C−H bond on the Cu−doped SBA−15 surface is 39.8 kcal/mol, and the Gibbs free energy change is 12.1 kcal/mol. Although this energy barrier is still relatively high, considering the energy released during the adsorption process (i.e., 32.6 kcal/mol), it is possible to overcome the energy barrier for the dissociation reaction, indicating that the dissociation processes of the C−H and C−S bonds of DMSO on the Cu−doped SBA−15 surface are competitive. A more detailed dissociation mechanism will be further discussed in future work, and the doping of other

transition metals and a variety of metals into the SBA−15 molecular sieve will be considered also in future work, to study their influence on the pyrolysis of oil shale.

**Author Contributions:** Conceptualization, X.M. and X.J.; methodology, L.L.; software, L.L.; validation, H.X. and Y.W.; formal analysis, X.M.; investigation, H.X.; resources, X.M.; data curation, H.X.; writing—original draft preparation, Y.W.; writing—review and editing, Y.W.; supervision, Y.W.; project administration, X.M.; funding acquisition, Y.W. All authors have read and agreed to the published version of the manuscript.

**Funding:** This research was funded by the National Oil Shale Research and Development Center Foundation of China, grant number 33550000-21-ZC0611-0013, National key research and development project, grant number2019YFA0705503, and Sinopec Science and Technology Project, grant number P20066.

**Data Availability Statement:** Not applicable.

**Conflicts of Interest:** The authors declare no conflict of interest.

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
