# Peer review of "Theoretical Investigation on the Catalytic Effect and Mechanism of Pure and Cu−Doped SBA−15 Molecular Sieves on the Decomposition of Dimethyl Sulfoxide"

_processes, doi:10.3390/pr11051386_

Round 1

Reviewer 1 Report

in this paper, the catalytic effect and mechanism of pure and Cu-doped SBA-15 molecular sieves on the decomposition of organic sulfur compounds in oil shale using density functional theory (DFT) method were explored, with DMSO serving as a model molecule for organic sulfur compounds in oil shale. The adsorption energies of DMSO molecules on both surfaces were found to be similar, but Cu-doped SBA-15 showed enhanced catalytic activity in dissociation reactions. On the pure SBA-15 surface, the Gibbs free energy changes for both possible reaction pathways of DMSO breaking C-S bonds were positive, and the activation energy barriers were high, suggesting that dissociation of C-S bonds in DMSO was unlikely to occur. On the Cu-doped SBA-15 surface, the Gibbs free energy change for the same reaction was negative, and the energy barrier was reduced by ~40 kcal/mol compared to that on the pure SBA-15 surface. These findings offer theoretical guidance for catalyst design and synthesis for in-situ pyrolysis of oil shale under suitable conditions.

The paper is well-written but needs some revisions.

1-      In the introduction, the applications of the proposed model should be mentioned in the beginning of the section.

2-      There are methods used to solve the proposed model that should be included.

3-      The motivation for this work should be added to the end of the introduction section.

4-      The parameters in the equations in section 2 should be defined clearly.

5-      Several typo errors should be revised.

6-      The conclusion section should contain the main findings of the paper.

7-      The references should be prepared according to the journal style.

8-      Authors should add some relevant papers in reference for a better presentation of the manuscript.

Reviewer 2 Report

I found your submitted work interesting and worth recommending for publication. 

Author Response

The reviewer found the manuscript  interesting and worth recommending for publication without providing any specific revision suggestions. Therefore, there was no response to the reviewer's comments.

Reviewer 3 Report

This work is worthwhile to be publish in this journal after minor revision. The following issues should be addressed:

1. Introduction is well-organized but the importance and novelty of the research should be highlighted and more clearly stated. The authors should give some examples of works in the bibliography, to clear the advantage of their work in comparison with those works.

2. The manuscript contains some minor typo/grammar errors, please check all of it.

3. Introduction part, if possible, some important and relative reports silica that could help:

Delta University Scientific Journal Vol.05-Iss.02 (2022) 321-339

https://doi.org/10.1016/j.jmrt.2022.03.067

https://doi.org/10.1016/j.colsurfa.2021.126361

https://doi.org/10.1007/s10971-022-05755-7

4. Abstract not targeted; the authors should rephrase it.

Hence, I recommend it accepted for publication after minor revisions.

Reviewer 4 Report

1

Abstract should contain more relevant research results, and link to the conclusion.

Conclusion should be improved, giving some more details of the research.

2

It seams that in some phrases, the author may want to use e.g. instead of i.e.

The abbreviation "e.g." stands for exempli gratia and means “for example.” 

The abbreviation "i.e." stands for id est and means “that is.”

3

Figure 5 and 6, to be adequated within the margin limits.

DFT should be explained.

4

Where the data come from? Please detail.

Where exactly was the experiments/tests carried out? Please detail.

How by means of which equipment/resources could the authors propose components strcuture adjustments? Details...

The graphs/figures, are being extracted from which tool?

5

I do not really understand this, and I would suggest the authors to rephrase.

"The transformation of oil shale into liquid shale oil"

Low by how much? Would be very interesting to add some quantitative values.

"the pyrolysis conversion rate of oil shale is relatively low"

6

I would be very carefull with such sentence. With technologies and developments, new reserves are being found, the recovery factor is being increased.

"As a result, the traditional energy reserves have proven inadequate to keep up with the swift pace of energy consumption brought forth by human development. "
